# The Mediating Effects of Innovativeness and System Usability on Students' Personality Differences: Recommendations for E-Learning Platforms in the Post-Pandemic Era

Mei-Hui Peng [1] and Bireswar Dutta [2,*]

1   Institute of Information Management, Minghsin University of Science and Technology, Hsinchu 300044, Taiwan
2   English Taught Program in Smart Service Management, Department of Information Technology and Management, Taipei Campus, Shih Chien University, Taipei 10462, Taiwan
*   Correspondence: bdutta67@gmail.com

**Abstract:** The COVID-19 pandemic forced higher education institutions to adopt e-learning systems to ensure continuous teaching and learning; however, this paradigm shift challenged students' learning processes and is considered unsuitable for continuous use. Thus, a model was developed and experimentally verified in the current study to determine the factors that influence students' uptaking of e-learning in the post-pandemic era. The Delphi method was employed to conceptualize the research framework, and structural equation modeling (SEM) was used to explore personality traits. The research model was then empirically tested by using data from 438 valid responses. The results showed that all personality traits, except for conscientiousness, significantly influenced the adoption of e-learning. The most decisive influencing trait was found to be extroversion ($r = 0.756$), whereas the trait that was found to have the most negligible impact was agreeableness ($r = 0.305$). Personal innovativeness and system usability were both found to highly correlate with a willingness to adopt e-learning. Except for the indirect effect of conscientiousness on the adoption of e-learning through system usability, all other personality traits were found to significantly mediate the adoption of e-learning through personal innovativeness and system usability. The results of this study could inspire stakeholders in the field of education, particularly e-learning platform designers, to consider students' personality traits and individual differences in the design of e-learning platforms, with the goal of increasing students' willingness and ability to adapt to these systems. The current study provides a contemporary perspective on the actions of e-learning users in the post-pandemic era.

**Keywords:** personality traits; post-pandemic era; sustainable; e-learning adoption; personal innovativeness; system usability; COVID-19 pandemic

## 1. Introduction

The COVID-19 pandemic presented a challenge for innovation and the development of efficient, cost-effective online platforms for student–teacher communication [1]. E-learning, a significant innovation in information technology (IT), encourages instant dialogue between students and instructors through the use of Internet-connected devices, such as smartphones, laptops, etc., whenever and wherever needed [2,3]. It offers a platform for exchanging knowledge with various audiences through audiovisual technologies, social network platforms, intranets, e-books, email, chat, blogs, and digital broadcasting networks [4,5]. The development of the Internet, along with the emergence of new low-cost technology and the COVID-19 pandemic [2], has brought significant attention to e-learning. This has drawn much interest from both business and academia because of its pivotal role, particularly during the pandemic [3,6].

The COVID-19 pandemic forced people to change how they behaved and socialized compared to before the pandemic [3,4]. It also forced educational institutions globally to

shift from face-to-face to online forms of education [7]. Almost every country forcibly closed educational institutions, directly influencing students' learning processes [2,4]. Higher educational institutions in Taiwan also switched to the delivery of online courses through a variety of e-learning platforms to guarantee the continuation of academic activities [3].

Due to its broad adoption, researchers started paying attention to e-learning in various sectors [7,8]. Additionally, the recent growth in research in this field can be attributed, in part, to the appeal among some scholars of studying both the online and offline components of e-learning [4,5]. Despite the intensity of interest in e-learning, there have been concerns about its uptake and utilization [9]. There is a lack of research on the motivational factors behind the adoption of e-learning, particularly in the post-pandemic era [1]. Because of the technological nature of e-learning, stakeholders must understand how users interpret and engage with e-learning systems, particularly during the pandemic. This information can help stakeholders to enhance the user experience, resulting in greater acceptance and usage [7].

It is crucial that today's academics understand students' intentions to use e-learning platforms [3,4]. It is difficult to convince students to use e-learning platforms that do not meet their expectations. Empirical studies have used a variety of theories, such as the unified theory of acceptance and use of technology (UTAUT), the theory of reasoned action (TRA), the theory of planned behavior (TPB), the model of personal computer use (MPCU), and the diffusion of innovation (DoI) theory, to explore the factors that influence the adoption of e-learning [10–12]. These theories have been used in both organizational and social contexts; however, few studies in the post-pandemic era have examined the effects of individual differences on students' intentions to adopt e-learning.

Personality traits are thought to affect students' behavioral intentions and manifest in their conduct [13,14]. The literature claims that a person's personality traits can significantly influence their behavior and ability to make decisions [13,14]. Additionally, a person's attitude is thought to correlate strongly with online teaching and learning.

Some recent studies have examined personality traits in the context of educational and learning environments, including online learning [7,9]; however, many of these studies neglected to consider how students' personalities influence their perceptions of a system's usefulness and how their creativity influences their intent to adopt e-learning in the post-pandemic era [9]. According to the literature, individuals open to innovation are more likely to engage with new technologies. People with different personalities will evaluate new systems differently, ultimately affecting how likely they are to use an e-learning platform in the post-pandemic era as opposed to before the pandemic [2,4]. To adequately address students' information system (IS) usage concerns, it is crucial to consider innovation and system usability challenges.

Based on this concise overview of the recent literature, we can identify a research gap in the relationship between students' psychological states and their creativity in utilizing e-learning platforms in the post-pandemic era. Furthermore, the impact of students' personality traits on their intention to accept a new system based on its usefulness has been overlooked. Thus, we suggest the following research questions to address this knowledge gap:

RQ1: Do students' individual differences impact their innovation regarding the adoption of e-learning in the post-pandemic era?

RQ2: Do students' individual differences impact system usability regarding the adoption of e-learning platform in the post-pandemic era?

RQ3: Do students' individual differences impact their adoption of e-learning in the post-pandemic era?

The current work intends to broaden the theoretical model of personality characteristics' applicability to post-pandemic-related circumstances, thereby contributing to the growing literature on e-learning. Thus, the aim of this study is to investigate the various motivations that affect the uptake of e-learning in post-pandemic environments.

## 2. Theoretical Background

### 2.1. Big Five Personality Traits

The five-factor model presented by McCrae and Costa [15] is currently the most widely used and acknowledged model, even though various extensively used models of personality traits continue to influence modern studies [16,17]. A critical fifth personality feature, neuroticism (or emotional stability), primarily related to the prediction of depressive and anxiety disorders, is included to explore the features more completely. The Big Five qualities are briefly described as follows in Table 1:

**Table 1.** Personality traits.

| Qualities | Expression |
|---|---|
| Extraversion | The degree to which a person interacts with the outside world and feels joy and other pleasant feelings. |
| Agreeability | The degree to which people respect moral principles like honesty and decency, social harmony, and teamwork. Amiable people typically view other people favorably. |
| Conscientiousness | The degree to which people place a premium on performance, value perseverance, and value planning. |
| Neuroticism | A tendency to respond emotionally and the extent to which people experience negative emotions. |
| Openness to experience | The degree to which an individual is self-aware, interested, and unconventional. |

The Big Five model was used in the current study to evaluate how particular personality traits influence students' behavioral intentions to adopt a new IS. Even previous studies adopted different adaptations for personality traits such as "talkative" for extraversion, "sympathetic" for agreeability, "disorganized" for conscientiousness, "temperamental" for neuroticism, and "imaginative" for openness to experience [14,16,17]. Peng and Dutta [14] explored how different personal characteristics influenced students' information privacy concerns, indirectly influencing their behavioral intentions to adopt an e-learning system during the pandemic era. Research in the literature employed theories such as the technology acceptance model, the theory of reasoned action, the theory of planned behavior, and the diffusion of innovation theory (DoI) to explore how individuals' personality differences influenced e-learning adoption intention [3,10,11,14,17,18]. Despite assertions that the Big Five personality traits are universal, the literature has questioned as to whether the Big Five are conceptually and methodologically incorrect, and findings from the prior studies have issues concerning their conceptual validity [13,19,20]. Thus, the current research explores the role of different personalities regarding adoption intentions in order to fill the gap, especially post-pandemic.

### 2.2. Personal Innovativeness

Innovation is considered to be "the extent to which a person is comparatively earlier in adopting an innovation than other members of his/her domain", where "comparatively earlier" denotes the actual adoption time as opposed to the perceived adoption time [21]. Strobl et al. [22] stated that an individual's innovativeness is a persistent trait or disposition that influences how a person interprets and reacts to innovations, with a high level of personal innovativeness leading to a more positive response. Individual innovativeness is commonly assessed concerning innovation diffusion, openness to novel systems [23], and desire to seek information from external entities [24]. The current study adopts a broader perspective on students' inventiveness, affecting their perception of the e-learning environment and their propensity to improvise as well as generate fresh ideas with which to tackle difficulties [25]. The impact of innovation on students' thoughts about implementing a new IS might fill in the gap as to how a student's innovativeness influences their behavioral intentions toward a novel IS, especially post-pandemic.

### 2.3. System Usability

Usability measures how user-friendly a user interface is in customers' interactions with interactive technology on a qualitative level [26]. To improve an interface for potential users, usability evaluation tasks primarily involve identifying system usability concerns [27]. Students claim that an interface's usability is the most crucial aspect of an e-learning platform when using it where high-level interactions occur [28]. Thus, usability is one of the most important criteria for evaluating the effectiveness of an e-learning platform's user interface [29].

The usability of an e-learning platform influences students' perceptions of their educational experiences; if an e-learning platform is challenging to use, students give up trying to use it. Research in the literature has assessed the usability of e-learning platforms to determine the relative value of the usability of design elements necessary in evaluating educational portals [28,30]. The relative importance of usability factors for an e-learning platform was explored by Alshehri et al. [31]. Muhammad et al. [32] looked at the relative importance of the design elements used in educational portals; however, the current study adopts a more comprehensive perspective on how students perceive the usability of e-learning platforms and highlights the significance of selecting user interface design concepts that are appropriate for various systems and contexts.

## 3. Hypotheses Development

### 3.1. System Usability and E-Learning Adoption Behaviors

Usability is the ability of a system to be utilized quickly and readily [27]. Usability is determined by several crucial factors, including learnability, effectiveness, memorability, error frequency, and subjective satisfaction [33]. Zaharias and Poylymenakou [30] claim that novelty, beauty, efficiency, reliability, perspicuity, and stimulation are qualities that have an impact on usability. Research in the literature has emphasized the importance of usability components in developing e-learning environments [27,30,33,34]. Scholtz et al. [34] discovered a strong correlation between system usability and e-learning adoption behaviors. The current study aims to understand how user interaction and gratification with an e-learning system are influenced by its usability.

**H1.** *The system usability of an e-learning platform positively influences e-learning adoption behaviors.*

### 3.2. Personal Innovativeness and E-Learning Adoption Behaviors

Personal innovativeness was rationally supported as significantly influencing adoption behaviors [22]. Strobl et al. [22] discovered that personal innovativeness indirectly affects self-efficacy in adopting an e-learning system. The work of Zheng et al. [35] indicated that personal innovativeness substantially impacts usage intention. Research in the literature validated relationships between technology and innovation, particularly regarding the employment of instructional technologies [35,36]. Individual innovativeness was defined by Al-Rahmi et al. [24] as the personal behaviors that are shown to be geared toward engaging in innovative activities at work. Individuals regard new sources of learning as easy to use, which will also influence their inclination to use them.

**H2.** *Personal innovativeness positively influences e-learning adoption behaviors.*

### 3.3. Personality Traits and E-Learning Adoption Behaviors

Research in the literature indicated that the adoption of training and occupational education positively correlates with an extroverted personality [37]. Extroverted pupils are motivated learners and usually do well in games [38]. They have expressive movements and are enthusiastic [38]. Extroverted students prefer to interact with others through IT technologies [39]. Therefore, extroverted students are more likely to benefit from and find e-learning systems engaging.

**H3a.** *Extroversion is positively associated with e-learning adoption behaviors.*

Openness to experience is the best indicator of the propensity for pursuing technical fields [40]. Due to the novelty, flair, and excitement of e-learning systems, students receptive to new experiences will likely be more engaged in them.

**H3b.** *Openness is positively associated with e-learning adoption behaviors.*

Neurotic persons are less willing to try changes or innovations; when they do, they are more likely to experience technology anxiety and irritation [41]. People with neuroses generally react unfavorably to new situations or changes, finding it difficult to relax and recognize their positives [16].

Neuroticism negatively impacts the adoption of training and occupational education [17]. No correlation between conscientiousness and the outcomes of the English tests was found in a survey on upper secondary schools in Germany, regardless of the other variables [17]. There is hardly any association between conscientiousness and the desire to accept new technology as a specialized area of attention, even though conscientiousness is occasionally seen as the most important of the Big Five in terms of success [42]. Thus, we propose the following:

**H3c.** *Neuroticism is negatively associated with e-learning adoption behaviors.*

**H3d.** *Conscientiousness is negatively associated with e-learning adoption behaviors.*

By being dependable, considerate, and averse to conflict, an agreeable person helps to create a positive learning environment [43]; they embrace contemporary technology [43]. Research in the literature has explored the idea that the best predictor of a person's specialization and preferences in mathematics, science, engineering, and technology, according to the literature, is agreeableness [44]; however, there was conflict between agreeableness and academic achievement for both sexes [45].

**H3e.** *Agreeableness is negatively associated with e-learning adoption behaviors.*

*3.4. Personality Traits and Personal Innovativeness*

Friendliness, aggression, and vigor are all characteristics of extraversion [19]. Research in the literature indicates that those with high extraversion levels more readily develop original ideas [16,17]. This enables extraverted individuals to actively participate in developing something new. It creates opportunities for utilizing and exploring knowledge, which is crucial for innovation [42]. Additionally, excitement and positive feelings motivate extroverted individuals to take risks [41]. According to Bühren and Steinberg [38], teams scoring higher on extraversion are likelier to do well when completing creative tasks.

**H4a.** *Extroversion is positively associated with personal innovativeness.*

Openness to experience has the most positive and well-supported impact on innovativeness. The research convincingly demonstrates that openness considerably benefits innovativeness in terms of creative capacity [42]. Intellectual curiosity, open-mindedness, inventiveness, and originality, as well as a variety of interests and information-seeking activities, define openness [45]. All of these inspire those with a high level of openness to challenge perceptions and try new things [19].

**H4b.** *Openness is positively associated with personal innovativeness.*

Knowledge of the impact of neuroticism on creativity is much more precise now. Coenen et al. [16] contend that people with highly neurotic personalities struggle to engage

in innovative behaviors and pursue innovative ideas because of the negative traits of anxiety, hostility, and self-consciousness [41], as well as the propensity to feel negative emotions [38]. Self-assurance and emotional stability, related to low levels of neuroticism, are characteristics innovators typically display [19,45].

**H4c.** *Neuroticism is negatively associated with personal innovativeness.*

There is conflicting data on how conscientiousness influences innovativeness, similar to the case of agreeableness. While conscientious peoples' planning-, organization-, and achievement-oriented tendencies sometimes stifle creative behavior [42], competence, tenacity, and self-control are necessary for the development of successful ideas [45]. Aligning with this idea, Rivers [17] found that conscientiousness positively affected a person's ability for invention. Hamilton et al. [41] found that high levels of conscientiousness significantly predict how well people accomplish creative tasks. The benefits of conscientiousness will eventually be needed to see creative ideas through to completion.

**H4d.** *Conscientiousness is negatively associated with personal innovativeness.*

Individual creativity has a tricky balance. While some personality attributes, such as adaptability, teamwork, and kindness, appear to foster invention [38], others, such as tolerance and conformity [19], may work against a person's propensity for innovation. As a result, it is not surprising that several studies have found no correlation between agreeableness and inventiveness [16,41]; however, it has been found that agreeability is a significant indicator of national-level innovation [42,45].

**H4e.** *Agreeableness is positively associated with personal innovativeness.*

### 3.5. Personality Traits and System Usability

Strong extraversion personalities typically attempt new things [16]. Research in the literature has shown that extraversion is positively connected with training and system satisfaction regarding system use [17]. The personality trait of extraversion has been considered as the ability to use new educational systems and technologies [45].

Different personality traits may have varying opinions on a system's usefulness. Neurotic learners are more likely to judge a new system's usability negatively. Conscientious learners are more likely to rate a new system's usability favorably than less conscientious ones [41]. Diligent students could see a system's value, which can help to improve their academic achievement [19]. Users' emotional evaluations of a system's usability was negatively impacted by neuroticism.

Openness to new experiences and agreeableness may substantially impact outcomes in terms of system usability. Similar personality traits, including agreeableness and openness to new experiences, may aid learners in determining the viability of a new system where their learning outcomes may be improved [45]. Agreeable students characteristically have a positive outlook and see their peers and technology as valuable partners [19]. Learning-supporting online tools are generally seen favorably by students who are open to new experiences [16].

**H5a.** *Extroversion is positively associated with the system usability of an e-learning platform.*

**H5b.** *Openness is significantly associated with the system usability of an e-learning platform.*

**H5c.** *Neuroticism is negatively associated with the system usability of an e-learning platform.*

**H5d.** *Consciousness is positively associated with the system usability of an e-learning platform.*

**H5e.** *Agreeableness is positively associated with the system usability of an e-learning platform.*

### 3.6. Mediating Pathways to E-Learning Adoption Behaviors

This study considered the mediating processes through which personality factors influence students' e-learning adoption behaviors. These mechanisms include individual creativity and system usability. We explore how these variables—personal innovativeness, system usability, and personality features—could mediate the relationship between personality traits and e-learning adoption behaviors. According to the study's anticipated method, personality factors influence adoption behavior levels by predicting personal ingenuity and system usability. As a result, we provide the following hypotheses:

**H6.** *Personal inventiveness will mediate between personality factors and e-learning adoption behaviors.*

**H7.** *System usability of the e-learning platform will mediate between personality factors and e-learning adoption behaviors.*

Based on the explanation above, the proposed study paradigm is shown in Figure 1.

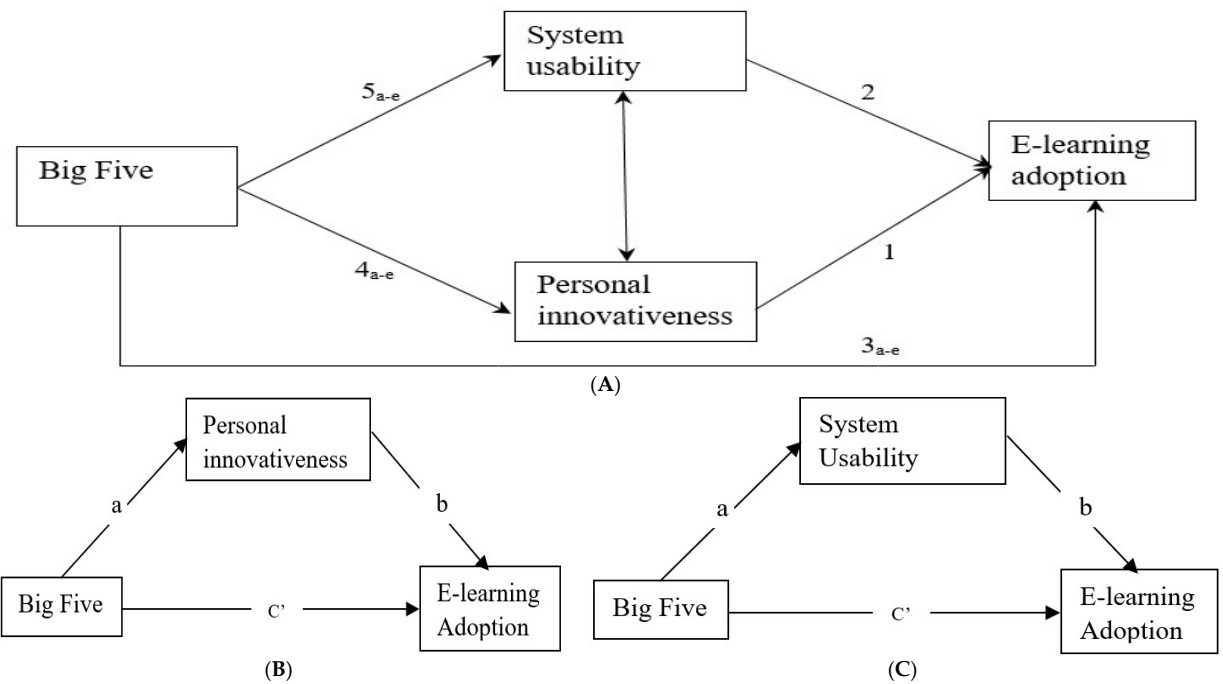

**Figure 1.** Research model. (**A**) Research model with the total effect. (**B**) Simple mediation model of personal innovativeness. (**C**) Simple mediation model of the system usability.

## 4. Materials and Methods

### 4.1. Research Setting and Data Collection

The rapid rise in COVID-19 cases forced Taiwan's colleges and universities to switch from traditional face-to-face instruction to online instruction. Thus, information was gathered by conducting an online survey. To clarify the study's goal, pre-defined criteria were employed in order to choose study participants: Firstly, the pupils should have a background in traditional classroom instruction in the first place. Second, students should have experience using digital devices such as computers, multimedia phones, iPads, etc., to demonstrate their fundamental understanding of ICT devices. Thirdly, students must spend at least 12 h each week using digital tools for education. The students who met the abovementioned requirements are considered contributors to the current investigation. Additionally, respondents were informed that they were free to stop participating in the study at any time.

The investigation was carried out within one month (from 2 December 2022 to 10 January 2023). The target demographic for this inquiry was students. We chose the convenience sampling approach because it is less expensive, has been used frequently in IS research, and enables researchers to obtain early data as well as look at trends without the inconveniences associated with using a randomized sample [3]. All of the participants were provided with informed permission forms and information sheets detailing the precise objectives of the current investigation [12,14]. An online survey was used to collect data. Additionally, participants were informed that they had the right to cancel their participation in the study at any time. The participants also received a brief explanation of how their data would be stored and the security measures implemented [14]. The following two elements influenced the decision to use this tactic. We first aim to fill in any knowledge gaps in data protection that may have caused participants to overstate the threat to their security. The second goal is to establish an appropriate level of security for using participant data.

The current research has not required institutional review board (IRB) approval as the participants were not asked to provide private information, such as their physical characteristics, genetic makeups, or psychological conditions. Additionally, no laboratory results were used. The participants were asked to fill out a questionnaire based on their knowledge and understanding of the advantages or disadvantages of adopting an e-learning platform during the post-pandemic period. Based on their knowledge, participants selected an option out of five (strongly disagree to strongly agree). Additionally, we briefly explained the overall operation of an e-learning system to our students. We did this for two reasons: First, to remove any participants' e-learning system apprehensions caused by the system's continual technological developments. Secondly, to form an accurate opinion about the system's potential future uses. Respondents were also informed about their privileges to be pulled out of participation during the survey.

Five hundred and forty-six people were contacted for data collection over eight weeks, and four hundred and forty-five survey questionnaires were returned. Of these, 12 were incomplete, leaving 438 useable replies (an effective response rate of 80%). The demographic data are presented in Table 2.

**Table 2.** Sample demographics.

| Item | Option | Count | Percentage % |
|---|---|---|---|
| Gender | Male | 227 | 51.76 |
| | Female | 211 | 48.24 |
| Age | 18–24 | 290 | 66.32 |
| | 25–30 | 129 | 29.38 |
| | >30 | 19 | 4.30 |
| Education level | Bachelors | 264 | 60.35 |
| | Associate degree | 126 | 28.77 |
| | Masters | 48 | 10.88 |

The distribution of the study population aligns with Taiwan's actual population [46]. Sixty percent of respondents came from undergraduate programs, compared to just ten percent from graduate programs. The rationale for this is that undergraduate programs enroll more students than graduate programs do.

### 4.2. Data Distribution

According to Hair et al. [47], if a dataset is smaller than 2000 entries then the Shapiro–Wilk test could be a better choice for testing data distribution. Since we have 438 valid responses, the Shapiro–Wilk test was used. Table 3 shows that the range of *p*-values is between 0.308 and 0.382. Thus, we can reject the alternative hypothesis and conclude that the data are from a normal distribution.

**Table 3.** Data distribution.

| | Shapiro–Wilk | | |
| | Statistic | df | Sig. |
|---|---|---|---|
| E-learning adoption | 0.946 | 426 | 0.326 |
| System usability | 0.886 | 445 | 0.308 |
| Personal innovativeness | 0.918 | 472 | 0.326 |
| Agreeableness | 0.854 | 484 | 0.342 |
| Openness to experience | 0.878 | 478 | 0.354 |
| Neuroticism | 0.934 | 462 | 0.362 |
| Conscientiousness | 0.949 | 418 | 0.376 |
| Extraversion | 0.968 | 486 | 0.382 |

### 4.3. Measurement of the Delphi Method

The current study developed and validated the proposed model by using mixed approaches. A literature review and in-depth interviews with subject matter experts from business and academia were part of the proposed model's development. The research instrument created for the study was then used to survey and test the study model experimentally. The responses from the empirical inquiry were further developed into a conclusive discussion and interpretation via focus group talks.

The basic conceptual framework was validated by using the Delphi methodology. A panel consisting of fifteen experts was established. Eight of the panel's fifteen experts were men who were Assistant Professors. Four of the panelists were female Assistant Professors. Three of the panelists were employed by a company that created and supported e-learning environments. Every panelist had over ten years of experience in their respective field and possessed a doctoral degree. The experts' average age ranged from 40 to 54 years old (nine belonged to the 40–52 group, and six belonged to the 45–54 group). Asking these specialists to serve as panelists raises severe concerns because their expertise is much more closely tied to educational information science. After two rounds of expert meetings, a research strategy and potential subjects for a pilot study were suggested. With a range of strongly disagree to strongly agree, we employed a five-point Likert scale. There were four-week gaps between each round of expert panel discussions to prevent the memory effect. The questionnaire was changed based on professional judgment and pertinent literature research to improve the content's validity. The final items employed for the current study are included in Table S1 (Supplementary Materials).

Each factor's mean value, standard deviation, and internal consistency were checked after receiving the questionnaire in the initial round from experts. Cronbach's alpha varied from 0.778 to 0.946 in this round. The expert panel suggested three changes: (1) making related and challenging-to-understand items more straightforward; (2) presenting the items in an organized manner; and (3) assigning one item to innovation. Due to the items' redundancy, two additional items were eliminated. Thirty-one items were thus sent for the second round of the Delphi procedure.

Following the second round, specific paths have a Cronbach's value of 0.852, while others have values between 0.881 and 0.992. Each factor's average relevance ranges from 3.89 to 5.00, with a standard deviation between 0.5 and 1.5. The proposed initial study framework's components of agreeableness, openness to experience, neuroticism, conscientiousness, extraversion, and adoption of e-learning remain unchanged in light of the comments made at the second-round discussion. Items for system usability were raised from 5 to 6, while those for individual inventiveness were increased from 3 to 4.

### 4.4. IRR Index

To test the rater agreement, the kappa statistic was executed. According to Hair et al. [47], a value (agreement) above 0.6 is reasonably acceptable. The current analysis shows that the level of agreement is 0.726 and statistically significant at the level of $p < 0.001$ (Table 4).

**Table 4.** Kappa analysis.

|  | Value | Asymp. Std. Error [a] | Approx. T [b] | Approx. Sig. |
|---|---|---|---|---|
| Measure of agreement kappa | 0.726 | 0.086 | 8.781 | 0.000 |
| N valid for cases | 438 |  |  |  |

[a]. Not assuming the null hypothesis. [b]. Using the asymptotic standard error, assuming the null hypothesis.

Cronbach's alpha ($\alpha$) was employed during each round of the Delphi method to bear out the internal consistency of the study items. It was also calculated as a measure of homogeneity for the ratings because intensifying uniformity was observed as a suggestion of agreement among the panelists.

Panelists were asked to provide their understanding, which ranged from 1 to 5. One is the least acceptable, and five is the most suitable. We then calculated the average value of each item for all panelists. This item was accepted if the average value was equal to or higher than 3. If the value was lower than 3 it was discarded. The kappa statistic and Cronbach's alpha ($\alpha$) were also evaluated to determine the item's inclusion for the final analysis.

*4.5. Procedure*

The current study's data analysis methods include reliability and validity testing, correlation analysis, and regression analysis. Data analysis was performed by using SPSS and AMOS, two statistical programs. Confirmatory factor analysis (CFA) is calculated to study the validity of measurement scales by AMOS. Cronbach's alpha and composite reliability (CR) are applied to examine the reliability by using SPSS. Data are analyzed using the structural equation modeling (SEM) technique to test the relationships in the conceptual research model, and the offered hypotheses are tested by using AMOS. SEM is used since it is a widespread technique in the social sciences and it eliminates observational errors from latent variable measurements [47]. Finally, Hayes' PROCESS macro was developed to test the mediating effect by using SPSS [48].

**5. Results**

*5.1. Descriptive Statistics and Correlation Analyses*

To validate the scales and assess the proposed research framework, an exploratory factor analysis (EFA) was conducted. The mean value and standard deviation (SD) of each measure, factor loadings, Cronbach's alpha, composite reliability, and average variance extracted (AVE) for each variable are given in Table 5.

As shown in Table 5, the Cronbach's alpha value for all of the factors exceeded the acceptable amount of 0.7 [47]. The factor loadings of all of the measures are higher than the recommended value of 0.7 [47].

Table 6 shows the correlations between e-learning adoption, personal innovativeness, system usability, and the five-factor personality traits.

Correlation analyses revealed that system usability had the most vital relationship with e-learning adoption (R = 0.876), which supported H1. Moreover, according to H2, personal innovativeness was positively associated with e-learning adoption, and it showed that a higher level of innovativeness contributed to adoption intention.

Among the personality traits, extroversion and openness were positively associated with e-learning adoption, supporting H3a and H3b. While neuroticism had a negative and significant relationship with e-learning adoption, supporting H3c, it was not significantly correlated with e-learning adoption. The findings showed that conscientiousness had a negative association with adoption intention; thus, H3d was not supported. Contrary to H3e, the results showed a positive and meaningful relationship between agreeableness and e-learning adoption behaviors; therefore, H3e was not supported.

**Table 5.** Mean, SD, and factor loadings of each measure.

| Construct | Item | Mean | SD | Factor Loadings | Cronbach's Alpha | Composite Reliability | AVE |
|---|---|---|---|---|---|---|---|
| E-learning adoption | BINT1 | 3.54 | 1.27 | 0.822 | 0.856 | 0.761 | 0.81 |
| | BINT2 | 3.15 | 1.23 | 0.786 | | | |
| | BINT3 | 3.08 | 1.25 | 0.814 | | | |
| | BINT4 | 3.52 | 1.21 | 0.894 | | | |
| System usability | SU1 | 3.08 | 1.37 | 0.816 | 0.868 | 0.752 | 0.83 |
| | SU2 | 3.33 | 1.24 | 0.839 | | | |
| | SU3 | 3.20 | 1.23 | 0.832 | | | |
| | SU4 | 3.34 | 1.38 | 0.856 | | | |
| | SU5 | 3.56 | 1.38 | 0.846 | | | |
| | SU6 | 3.14 | 1.24 | 0.876 | | | |
| Personal innovativeness | PI1 | 3.19 | 1.45 | 0.904 | 0.852 | 0.862 | 0.78 |
| | PI2 | 3.10 | 1.30 | 0.842 | | | |
| | PI3 | 3.63 | 1.31 | 0.804 | | | |
| | PI4 | 3.56 | 1.35 | 0.917 | | | |
| Agreeableness | AGR1 | 3.54 | 1.29 | 0.928 | 0.882 | 0.841 | 0.72 |
| | AGR2 | 3.11 | 1.40 | 0.883 | | | |
| | AGR3 | 3.09 | 1.26 | 0.840 | | | |
| Openness to experience | OPE1 | 3.25 | 1.29 | 0.916 | 0.812 | 0.765 | 0.80 |
| | OPE2 | 3.55 | 1.34 | 0.882 | | | |
| | OPE3 | 3.57 | 1.31 | 0.783 | | | |
| | OPE4 | 3.51 | 1.34 | 0.894 | | | |
| | OPE5 | 3.46 | 1.27 | 0.821 | | | |
| Neuroticism | NEUR1 | 3.59 | 1.34 | 0.926 | 0.850 | 0.817 | 0.76 |
| | NEUR2 | 3.43 | 1.36 | 0.829 | | | |
| | NEUR3 | 3.04 | 1.22 | 0.812 | | | |
| Conscientiousness | CNS1 | 3.02 | 1.29 | 0.818 | 0.875 | 0.782 | 0.74 |
| | CNS2 | 3.40 | 1.31 | 0.813 | | | |
| | CNS3 | 3.08 | 1.25 | 0.912 | | | |
| | CNS4 | 3.15 | 1.28 | 0.860 | | | |
| Extraversion | EXT1 | 3.05 | 1.25 | 0.825 | 0.892 | 0.879 | 0.80 |
| | EXT2 | 3.27 | 1.21 | 0.918 | | | |
| | EXT3 | 3.43 | 1.35 | 0.823 | | | |
| | EXT4 | 3.00 | 1.28 | 0.835 | | | |

Extroversion, openness, and agreeableness were positively and significantly associated with personal innovativeness; thus, H4a, H4b, and H4e were supported. Additionally, neuroticism and conscientiousness were negatively associated with personal innovativeness; therefore, H4c and H4d were also supported.

Extroversion had the most substantial positive relationship with system usability; thus, H5a was supported. Openness was significantly and positively related with system usability, supporting H5b. Similarly, agreeableness was positively, yet less strongly, associated with system usability, supporting H5e. The findings also showed that neuroticism was negatively correlated with system usability; thus, H5c was supported. H5d was not supported because no significant relationship was found between conscientiousness and system usability.

The variance inflation factor (VIF) test was explored to test multicollinearity. According to Hair et al. [47], values higher than 10 indicate a high VIF. Table 6 shows that the VIF of each construct was much lower than the suggested value, ranging from 2.137 to 3.258; thus, we can conclude that no multicollinearity was seen.

### 5.2. Reliability and Convergent Validity

Reliability was verified through Cronbach's alpha and composite reliability (CR) to measure the model's internal consistency. The Cronbach's alpha and CR value of each construct were higher than the recommended value of 0.70 [47], ranging from 0.805 to 0.878 and 0.938 to 0.981 (Table 5), inferring appropriate reliability and consistency. The convergent validity of the scales was explored by using three standards suggested by Hair et al. [47]: (1) the loadings of each indicator should be higher than 0.70; (2) the CR value should exceed 0.70; and (3) the average variance extracted (AVE) should be higher than 0.50. As Table 5 endorses, the factor loading of each item is well above 0.70. CR values

have ranged from 0.72 to 0.83 (Table 5). The AVE value of constructs ranged from 0.72 to 0.83; thus, each condition for convergent validity is met (Table 5).

### 5.3. Discriminant Validity

To test discriminant validity, Hair et al. [47] suggested that the square root of the AVE of the construct should be higher than the estimated correlation shared between the construct and other constructs in the model. Table 7 shows that the square root of the AVE for each construct was more significant than the correlation values of the construct, thus meeting the condition for discriminant validity.

### 5.4. Regression Analysis

A regression analysis was conducted to test the conceptual research model's relationships. It is a consistent method of determining which factors matter the most, which factors can be ignored, and how they influence each other regarding students' e-learning adoption intentions during the post-pandemic period. To achieve the purpose of the current study, ordinal logistic regression (ordinal regression) is used to predict students' e-learning adoption intentions during the post-pandemic period based on independent factors such as personality traits, personal innovativeness, and system usability. The results are shown in Table 8.

The common method bias is controlled by including the method factor in the theoretical model in a way reflected by the items from the primary constructs and their measures.

The bootstrapping strategy, which resamples a single dataset to produce numerous simulated samples, is employed because it is an efficient alternative; moreover, it enables the creation of confidence intervals, standard errors, and hypothesis testing.

By reducing the bootstrap estimates of the prediction error, which are produced depending on the size of the dataset, the bootstrap model selection technique is used to choose the subset of variables.

The results showed that all of the personality traits significantly influenced innovativeness at the 0.001 level; however, conscientiousness was influenced considerably at a significance level of 0.05. The most effective trait was openness ($\beta$ = 0.592; $R^2$ = 0.351), and the least effective one was conscientiousness ($\beta$ = −0.112; $R^2$ = 0.013). Extroversion, openness, and agreeableness were positively associated with innovativeness, while neuroticism and conscientiousness had a negative impact on innovativeness (Figure 2).

All of the personality traits significantly affected the system usability of the e-learning platform, except conscientiousness ($\beta$ = −0.074; t-value = −1.275). Among the personality traits, extroversion had the most significant impact on system usability ($\beta$ = 0.885; $R^2$ = 0.617), while agreeableness had the least significant effect ($\beta$ = 0.305; $R^2$ = 0.09). Extroversion, openness, and agreeableness significantly impacted system usability, while neuroticism had a negative impact ($\beta$ = −0.424; $R^2$ = 0.187). Similarly, personal innovativeness positively impacted the system usability of the e-learning platform ($\beta$ = 0.737; $R^2$ = 0.526).

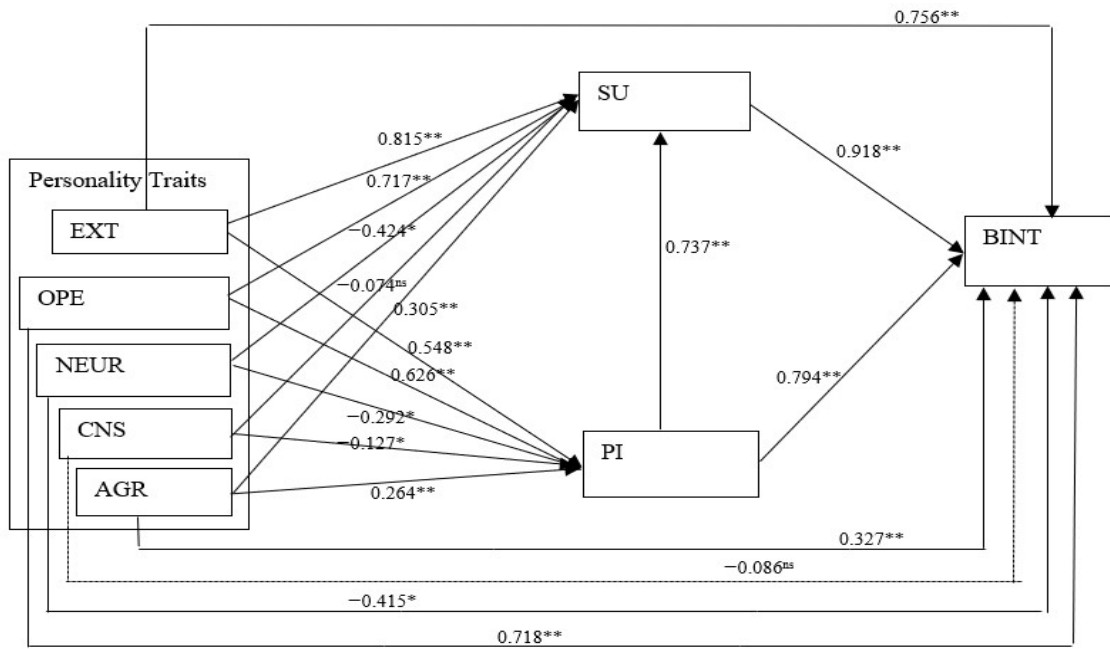

EXT= Extraversion; OPE= Openness to experience; NEUR= Neuroticism; CNS= Conscientiousness; AGR= Agreeableness; PI= Personal innovativeness; SU= System usability; BINT= E-learning adoption

**Figure 2.** Results of the regression analysis. ns = Not supported; * $p < 0.05$; ** $p < 0.01$.

All of the personality traits, except conscientiousness, significantly affected e-learning adoption behaviors ($\beta = -0.086$; t-value = −1.417). Extroversion was the most effective trait ($\beta = 0.756$; $R^2 = 0.568$), while agreeableness was the least effective trait ($\beta = 0.327$; $R^2 = 0.094$) on e-learning adoption behaviors. Extroversion, openness, and agreeableness positively impacted e-learning adoption behaviors, while neuroticism ($\beta = -0.415$; $R^2 = 0.194$) had a significantly negative effect; conscientiousness had a negative but insignificant impact on e-learning adoption behaviors.

Personal innovativeness ($\beta = 0.794$; $R^2 = 0.482$) and system usability ($\beta = 0.918$; $R^2 = 0.853$) positively impacted e-learning adoption behaviors. Incontrovertibly, system usability was more effective on e-learning adoption behaviors than personal innovativeness.

Finally, H6 and H7 were proposed to determine relationships between personality traits, and e-learning adoption behaviors were meditated by personal innovativeness and system usability. Model 4 (simple mediation) of the Hayes process is used for the current analysis. The results of a bootstrapping mediation analysis for all of the personality traits are presented in Table 9.

The bootstrapping analysis indicated that the mediating role of personal innovativeness in the relationship between extroversion ($\beta = 0.1527$, SE = 0.0226, and CI = [0.1115, 0.1921]), openness ($\beta = 0.1942$, SE = 0.0223, and CI = [0.1276, 0.2478]), neuroticism ($\beta = -0.1237$, SE = 0.0217, and CI = [−0.1206, −0.0947]), conscientiousness ($\beta = -0.0667$, SE = 0.0233, and CI = [−0.1023, −0.0216]), agreeableness ($\beta = 0.0919$, SE = 0.0219, and CI = [0.0825, 0.1345]), and e-learning adoption behavior is supported. The findings also demonstrated that the system usability of the e-learning platform has a mediating role in the relationship between extroversion ($\beta = 0.4551$, SE = 0.0268, and CI = [0.4118, 0.5032]), openness ($\beta = 0.4162$, SE = 0.0279, and CI = [0.3015, 0.4226]), neuroticism ($\beta = -0.2747$, SE = 0.0315, and CI = [−0.3018, −0.2226]), agreeableness ($\beta = 0.1597$, SE = 0.0304, and CI = [0.0976, 0.2234]), and e-learning adoption behaviors. Principally, the indirect effects of all personality traits on e-learning adoption through personal innovativeness and system usability are significant, except the indirect impact of conscientiousness on e-learning adoption through system usability.

**Table 6.** Results of correlation analyses among the main variables.

| Construct | E-Learning Adoption | System Usability | Personal Innovativeness | Agreeableness | Openness to Experience | Neuroticism | Conscientiousness | Extraversion | VIF |
|---|---|---|---|---|---|---|---|---|---|
| [a] E-learning adoption | 1 | | | | | | | | |
| SU | 0.876 ** | 1 | | | | | | | 2.268 |
| PI | 0.765 ** | 0.746 ** | 1 | | | | | | 2.624 |
| EXT | 0.793 ** | 0.769 ** | 0.531 ** | 1 | | | | | 2.186 |
| OPE | 0.742 ** | 0.676 ** | 0.566 ** | 0.568 ** | 1 | | | | 3.258 |
| NEU | −0.412 ** | −0.363 ** | −0.281 ** | −0.327 ** | −0.256 ** | 1 | | | 2.137 |
| CNS | −0.067 | −0.052 | −0.158 * | −0.071 | −0.027 | 0.326 ** | 1 | | 3.156 |
| AGR | 0.385 ** | 0.241 ** | 0.172 | 0.494 ** | 0.512 ** | −0.076 | 0.028 | 1 | 2.870 |

[a] Dependent variable: E-learning adoption; EXT = extraversion; OPE = openness to experience; NEUR = neuroticism; CNS = conscientiousness; AGR = agreeableness; PI = personal innovativeness; and SU = system usability. $* p < 0.05$; $** p < 0.01$.

**Table 7.** Average variance extracted and discriminant validity.

| Construct | E-Learning Adoption | System Usability | Personal Innovativeness | Agreeableness | Openness to Experience | Neuroticism | Conscientiousness | Extraversion |
|---|---|---|---|---|---|---|---|---|
| E-learning adoption | 0.90 | | | | | | | |
| System usability | 0.87 ** | 0.91 | | | | | | |
| Personal innovativeness | 0.76 ** | 0.74 ** | 0.88 | | | | | |
| Extroversion | 0.79 ** | 0.76 ** | 0.53 ** | 0.84 | | | | |
| Openness | 0.74 ** | 0.67 ** | 0.56 ** | 0.56 * | 0.89 | | | |
| Neuroticism | −0.41 * | −0.36 * | −0.28 ** | −0.32 * | −0.25 * | 0.87 | | |
| Conscientiousness | −0.06 | −0.05 | −0.15 * | −0.07 | −0.02 | 0.32 * | 0.86 | |
| Extraversion | 0.38 * | 0.24 * | 0.17 | 0.49 * | 0.51 ** | −0.07 | 0.02 | 0.89 |

$* p < 0.05$; $** p < 0.01$.

**Table 8.** Regression analysis results.

| DV / IV | R² | Personal Innovativeness β (*p*-Value) | t-Value | R² | System Usability β (*p*-Value) | t-Value | R² | E-Learning Adoption β (*p*-Value) | t-Value |
|---|---|---|---|---|---|---|---|---|---|
| EXT | 0.376 | 0.548 ** | 11.617 | 0.617 | 0.815 ** | 25.119 | 0.568 | 0.756 ** | 20.937 |
| OPE | 0.462 | 0.626 ** | 14.246 | 0.483 | 0.717 ** | 19.576 | 0.557 | 0.718 ** | 18.464 |
| NEUR | 0.086 | −0.292 ** | −6.187 | 0.187 | −0.424 ** | −9.360 | 0.194 | −0.415 ** | −8.557 |
| CNS | 0.027 | −0.127 * (0.01) | −3.571 | 0.007 | −0.074 (0.317) | −1.275 | 0.007 | −0.086 (0.31) | −1.417 |
| AGR | 0.056 | 0.264 ** | 5.668 | 0.09 | 0.305 ** | 5.852 | 0.094 | 0.327 ** | 6.351 |
| PI | | | | 0.526 | 0.737 ** | 21.127 | 0.482 | 0.794 ** | 19.27 |
| SU | | | | | | | 0.853 | 0.918 ** | 43.175 |

EXT = extraversion; OPE = openness to experience; NEUR = neuroticism; CNS = conscientiousness; AGR = agreeableness; PI = personal innovativeness; SU = system usability; and BINT = E-learning adoption. $* p < 0.05$; $** p < 0.01$.

**Table 9.** Indirect effects of personality traits on e-learning adoption through personal innovativeness and system usability.

| Indirect Path | β | SE | 95% Bootstrap CI Lower Limit | Upper Limit | Results |
|---|---|---|---|---|---|
| Extroversion → personal innovativeness → E-learning adoption | 0.1527 | 0.0226 | 0.1115 | 0.1921 | Supported |
| Openness → personal innovativeness → E-learning adoption | 0.1942 | 0.0223 | 0.1276 | 0.2478 | Supported |
| Neuroticism → personal innovativeness → E-learning adoption | −0.1237 | 0.0217 | −0.1206 | −0.0947 | Supported |
| Conscientiousness → personal innovativeness → E-learning adoption | −0.0667 | 0.0233 | −0.1023 | −0.0216 | Supported |
| Agreeableness → personal innovativeness → E-learning adoption | 0.0919 | 0.0219 | 0.0825 | 0.1345 | Supported |
| Extroversion → system usability → E-learning adoption | 0.4551 | 0.0268 | 0.4118 | 0.5032 | Supported |
| Openness → system usability → E-learning adoption | 0.4162 | 0.0279 | 0.3015 | 0.4226 | Supported |
| Neuroticism → system usability → E-learning adoption | −0.2747 | 0.0315 | −0.3018 | −0.2226 | Supported |
| Conscientiousness → system usability → E-learning adoption | −0.0418 | 0.0341 | −0.1126 | 0.0197 | Rejected |
| Agreeableness → system usability → E-learning adoption | 0.1597 | 0.0304 | 0.0976 | 0.2234 | Supported |

## 6. Discussion

One of the critical elements in creating a better aim to investigate a new IS is innovation. The current study found a strong correlation between individual inventiveness and the uptake of e-learning, which is consistent with previous research [41]. According to the findings, students believe that sustainable e-learning is more advantageous for their learning processes if a system has explicit, informed, consistent, intelligible, and well-formatted course content. This, in turn, increases their desire to embrace a sustainable IS.

The results of the present study investigated the relationship between system usability and adoption intention. When an e-learning system has a well-designed interface, an ideal layout, effective navigation, and quick answers, students may use it for learning. To increase students' willingness to use e-learning institutions should provide better communication and real-time technical support, in addition to distributing study materials more reliably and safely.

More extraverted (better creativity, self-assurance, and self-sufficiency in their competency to function and act) students determine passionate confidence in their ability to choose an e-learning system. Students with high agreeableness scores are more inclined and trustful than less agreeable students; they are less interested in and scared by the prospect of adopting a new IT system, such as e-learning. It is rational to presume that less agreeable students are more critical and skeptical, which could hinder them from taking advantage of the new systems, even in the post-pandemic period. Neurotic pupils worry about potential bad outcomes, which negatively impacts overall adoption intention [49]. As people use their thinking to understand the context and are ready to receive new opportunities in the post-pandemic era, openness to experience is positively and significantly associated with perceptions of adoption; however, a link between conscientiousness and pupils' adoption behaviors is unfavorable [50]. Students with high conscientiousness can better control themselves, so they tend to avoid dangerous behaviors. More conscientious students tend to avoid risks to a greater degree. These findings align with those of Peng and Dutta [14]; however, the inferences need to be studied further, as limited studies have evaluated the significance of personality characteristics concerning educational IT adoption in the post-pandemic era [3].

The current study can be seen as a first step in determining how students' psychological conditions affect their decisions to accept new e-learning platforms. Additionally, the research instrument offers a comprehensive assessment of sustainable e-learning platforms from users' viewpoints in the post-pandemic era (with technical, behavioral, or user personality variations).

A key component of student innovation is agreeableness, as explored by Hamilton et al. [41,42]. The study indicates that extraversion, agreeableness, and openness to experience have a significant positive association with individual innovativeness, but that neuroticism and conscientiousness have a significant negative association with innovativeness. These results align with the existing research [41]. These results show that extraverted pupils can better complete creative tasks [38]. Students with a high level of openness can undertake a creative assignment with determination. Students with highly neurotic personalities may

find it challenging to display innovative behaviors and explore innovative ideas, according to research on the adverse effects of neuroticism on students' capacity for innovation [19]. The study proves that being highly controversial is a reliable indicator of an individual's capacity to carry out inventive tasks [42].

Our analysis identified extroversion as the most influential trait on the system usability of the e-learning platform metrics. It indicated that extraverted students generally took more time to adopt a new IS than introverted persons, who were more successful; however, they were more prone to share their experiences of using the system, which indirectly helps in adoption. More agreeable students tended to rate the e-learning system better, since those students were friendlier by nature. The trait of conscientiousness showed an insignificant correlation with the system usability of the e-learning platform. This finding is surprising since conscientiousness is considered to be one of the most influencing factors for interpreting students' behaviors in other essential areas [45]. Since students with higher conscientiousness are more dutiful and accurate, we assumed that they might possess better consideration and, thus, be less critical concerning the e-learning system. Additional research is required to come to a decision. The negative correlation of neuroticism indicates that the lower their value of neuroticism, the less likely students are willing to adopt the system. The study's findings suggest that students who score high in openness are eager to explore more options and prefer situations that provide variety while solving problems in the post-pandemic era [16]. They are willing to explore the features associated with systems and services.

Personal innovativeness and the system usability of the e-learning platform mediated the relationship between personality traits and the adoption of sustainable e-learning; however, not all personality traits proposed by the Big Five model positively affected the system usability of the e-learning platform. Extroversion, openness, and agreeableness positively affected system usability, while neuroticism had a negative impact. Previous studies also indicated a positive relationship between extroversion [29], openness [30], agreeableness [31], and system usability. Research in the literature reported a positive relationship between extroversion [32], agreeableness [33], and system usability. The findings of this study urge the more effective utilization of technology. Before spending more money on new IT installations, governments and providers of e-learning services should consider these facts.

Regarding the negative correlation between neuroticism and the system usability of the e-learning platform, Zaharias and Poylymenakou [30] demonstrated how neuroticism predicted increases in negative effects after a stressful condition due to the pandemic. More extroverted students are socially dominant and endorse hierarchy, which would lead them to be more cautious about selecting a new system. Since affective commitment is a "double-edged sword that predisposes individuals to feel both bad and the good more profoundly, and therefore, it is difficult to explain its influence on innovativeness [16]", it is impossible to link students' openness to experience to affective commitment. Therefore, it is advised to research openness in various statistical populations. Stakeholders may develop strategies and implement designs that will motivate students to use the system by having a complete understanding of the elements impacting the use of sustainable e-learning. The system should be designed with various original and beautiful interfaces.

## 7. Conclusions

The ambition of students to adopt IT is one of the essential aspects of educational IT adoption. E-learning is considered to be one of the most secure and safe ways to provide lectures and better communication with instructors and students, particularly in the post-pandemic era. The current study examined these interactions to better understand how students' personality traits influence innovativeness and system usability, which affect students' intentions to use e-learning systems. For the primary group of sustainable e-learning system users, students' goals are crucial for ensuring that the anticipated benefits will be fulfilled.

The SEM analysis showed that the model was more accurate and robust in explaining why students were more likely to use a sustainable e-learning system post-pandemic. According to the current study, students' intentions to adopt sustainable e-learning was favorably influenced by extroversion, agreeableness, and openness to new experiences. Neuroticism and scrupulosity have a negative impact. The system's user-friendliness and individual ingenuity mediated the relationship between personality traits and the adoption of e-learning. Extroversion, openness to new experiences, and agreeableness all significantly impacted system usability, but neuroticism had the reverse impact. The negative association between neuroticism and system usability reinforced the assumption that the pandemic caused neuroticism to be more negative; however, studies reveal that extroverted people are more particular when selecting a new system.

## 8. Contributions

### 8.1. Academic Implications

One of the earliest studies to examine personality qualities and how they affect students' capacity for innovation, their ability to use systems with sustainable e-learning teaching styles, and their plans to use sustainable e-learning platforms is presented here. The results of this study significantly expand the corpus of existing theoretical knowledge. We have added fresh perspectives to the body of literature by fusing students' Big Five personality traits with the setting of sustainable e-learning teaching technologies in the post-pandemic era. Earlier studies that employed personality factors as predictors of students' behaviors have been enhanced and redirected by our team [3,4,11]. Furthermore, we have provided educational psychiatrists with a fresh perspective on how students' personalities changed throughout COVID-19 and how they feel about as well as act in response to innovation and e-learning platforms. Finally, considering that the current study's main focus was on how e-learning environments were adopted in the post-pandemic era, any advancement in our understanding of particular phenomena may lead to a higher level of the post-implementation adoption of educational technology.

### 8.2. Practical Implications

The current study offers information that e-learning providers and decision makers may use to comprehend students' perspectives and retain their enthusiasm for using e-learning systems as tools for knowledge and learning. Universities can create plans and devise strategies as well as learning platforms to improve their understanding of ICT. E-learning systems should be designed with various unique traits and displays. If lecturers and administrators provide the necessary support and direction, students will be encouraged to adopt e-learning for instructional purposes. Using an e-learning platform effectively in the post-pandemic era would create an environment that supports autonomy and improves learning [12]. Second, if COVID-19 continues to be a major factor, the data will allow policymakers to decide whether to keep offering online education or switch to a hybrid approach. We advise policymakers to concentrate on a teaching method that involves students receiving online instruction while actively participating in real-world tasks. The problems that followed the coronavirus outbreak demonstrate how sustainable e-learning can raise educational standards while increasing financial efficiency. By utilizing fewer resources, e-learning enables higher academic levels. Finally, this study's findings helped us to better understand how students' personality traits affected their decision to enroll in an online course in the post-pandemic era. Investigating outcomes to enhance students' competency, personality, and self-sufficiency via educational IT service providers will help produce effective transitions for students' learning behaviors.

## 9. Limitations and Suggestions for Future Studies

This is one of the first studies to provide empirical evidence of the influence of personality traits, system usability, and individual creativity on students' acceptance of sustainable e-learning in the post-pandemic era. Even though the study's findings are intriguing, cer-

tain limitations require more investigation. Students from Taiwan made up the initial sample size. Therefore, evaluating the conceptual model's causal relationships in academic settings in other countries with diverse cultural traditions would be preferable. The study also examines the variables that affect how readily students adopt sustainable e-learning. It is suggested that future studies investigate the variables that affect instructors' adoption of e-learning and contrast their findings with those from the current study.

**Supplementary Materials:** The following supporting information can be downloaded at: https://www.mdpi.com/article/10.3390/su15075867/s1, Table S1: Constructs and items. The questionnaire used for the current study is also included. Reference [51] is cited in the Supplementary Materials.

**Author Contributions:** Data curation, B.D. and M.-H.P.; formal analysis, B.D.; investigation, B.D.; methodology, B.D. and M.-H.P.; resources, M.-H.P.; software, M.-H.P.; validation, B.D. and M.-H.P.; writing—review and editing, original draft, B.D. All authors have read and agreed to the published version of the manuscript.

**Funding:** This research received no external funding.

**Institutional Review Board Statement:** Not applicable.

**Informed Consent Statement:** Not applicable.

**Data Availability Statement:** All data are available within the published paper, and the questionnaire used for the current study is included in the supplementary document.

**Conflicts of Interest:** The authors declare no conflict of interest.

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
