# Peer review of "The Mediating Effects of Innovativeness and System Usability on Students’ Personality Differences: Recommendations for E-Learning Platforms in the Post-Pandemic Era"

_sustainability, doi:10.3390/su15075867_

Round 1

Reviewer 1 Report

Dear Authors,

I really enjoyed reading your manuscript. I have left my comments in the attached file. Please carefully apply them as much as you can.

Author Response

Reviewer 1

  1. The title seems ungrammatical! The Mediating Effects of Students’ Innovativeness and System Usability on their Personality Differences: The Recommendation in the E-learning Platform at Post-Pandemic Time.

Ans. Thank you for your suggestion. I modify the Title, which will be found on Line 2-4.

The Mediating Effects of Students’ Innovativeness…at Post-Pandemic Time Line 2-4.

  1. It has to be moved to immediately after “Structural equation modeling”.

Ans. Thank you for your suggestion. I modify the sentence, which will be found on Line 16.

  1. What are these numbers for?

Ans. Thank you for your suggestion. I modify the sentence, which will be found on Line 18.

  1. Ditto

Ans. Thank you for your suggestion. I modify the sentence, which will be found on Line 19.

  1. Ditto

Ans. Thank you for your suggestion. I modify the sentence, which will be found on Line 21.

  1. Ditto

Ans. Thank you for your suggestion. I modify the sentence, which will be found on Line 22.

  1. You should either write the number or the scholar’s name.

Ans. Thank you for your suggestion. I modify the sentence, which will be found on page 1.

It offers a platform for exchanging…digital broadcasting networks [4,5].

  1. It is good that you have cited different scholars to base your work, but this is the introduction part! You should also try to include your own words, justify the need for your work, and say what it is important to do this research.

Ans. Thank you for your suggestion. I modify the sentence, which will be found on page 1.

E-learning has received significant notice…role, particularly during the epidemic [3,6].

  1. Ungrammatical

Ans. Thank you for your suggestion. I modify the sentence, which will be found on Line 44-45.

The COVID-19 pandemic forced humans to…socialized before the pandemic [3,4]. Line 44-45

  1. Here citing a work is needed.

Ans. Thank you for your suggestion. I include the references, which will be found on Line 70-72.

The literature claims that a person's personality…ability for decision-making [13,14]. Line 70-72

  1. More likely

Ans. Thank you for your suggestion. I modify the sentence, which will be found on Line 78-79.

According to the literature, individuals…their intention to try new technology. Line 78-79

  1. Why? I think you should rephrase yourself. Thus, this study aims to ……………

Ans. Thank you for your suggestion. I modify the sentence, which will be found on Line 78-79.

Thus, the aim of the study is to investigate…pandemic-inspired environments. Line 98-100

  1. Stick to the journal’s way of citation.

Ans. Thank you for your suggestion. I modify the sentence, which will be found on Line 103-105.

The Five-Factor Model presented by McCrae and Costa [15]…studies [16,17]. Line 103-105

  1. Or comparatively earlier??

Ans. Thank you for your suggestion. I modify the sentence, which will be found on Line 128-130.

Innovation is considered "the extent to which…perceived adoption time [21]. Line 128-130

  1. Take care of such similar cases based on the Journal, please.

Ans. Thank you for your suggestion. I modify the sentence, which will be found on Line 153-154.

The relative importance of usability factors…explored by Alshehri et al. [31]. Line 153-154

  1. Ungrammatical !

Ans. Thank you for your suggestion. I modify the sentence, which will be found on Line 351.

4.3. Measurement of the Delphi method Line 351

  1. A run-on sentence! Please check the manuscript for similar cases.

Ans. Thank you for your suggestion. I modify the sentence, which will be found on Line 361-362.

Every panelist has over ten years of experience… and receives doctoral degrees. Line 361-362

  1. Stay consistent regarding the tenses.

Ans. Thank you for your suggestion. I modify the sentence, which will be found on Line 361-362.

Every panelist has over ten years of experience… and receives doctoral degrees. Line 361-362

  1. Five

Ans. Thank you for your suggestion. I modify the sentence, which will be found on Line 366-367.

With a range of strongly disagree to strongly…employed a five-point Likert scale. Line 366-367

  1. I didn’t see a Table under such a name!

Ans. Thank you for your suggestion. I include the Table, in Appendix.

  1. I just found it on another attached file. Maybe it is better you say: Table A1 (appendix A)

Ans. Thank you for your suggestion. I modify the sentence, which will be found on Line 371-372.

The final items employed for the current…included in Table A1 (Appendix A). Line 371-372

  1. The length of your sentences here does not match that of the previous parts, showing that the other parts have been mainly paraphrases from others. Please try to combine your sentences here too.

Ans. Thank you for your suggestion. I modify the sentence, which will be found on Line 370-375.

The final items employed for the current…from 0.778 to 0.946 in this round. Line 370-375

  1. A great point you have mentioned here. Bravo.

Ans. Thank you for your suggestion.

  1. It is

Ans. Thank you for your suggestion. I modify the sentence, which will be found on Line 411-413.

SEM is used since it is a widespread technique…variable measurement [47]. Line 411-413

  1. It is better to connect the two sentences together. Some of your sentences are too short, while some others are too long. Stay in between, please.

Ans. Thank you for your suggestion. I modify the sentence, which will be found on Line 452-454.

Thus, H5c was supported. H5d was not supported…and system usability. Line 452-454

  1. The way you have reported the results is really good, but your paragraphing is weak.

Ans. Thank you for your suggestion. I modify the sentence, which will be found on Line 523-524.

Model 4 (Simple mediation) of the Hayes…used for the current analysis. Line 523-524

  1. I highly recommend moving the Tables to the part where you have discussed their content. Two Tables or Figures should not be put one after another without any text. Please kindly consider this point.

Ans. Thank you for your suggestion. I rearrange the table, which will be found throughout the paper.

  1. The

Ans. Thank you for your suggestion. I modify the sentence, which will be found on Line 479-480.

Regression analysis was conducted…conceptual research model's relationships. Line 479-480

  1. Not needed again.

Ans. Thank you for your suggestion. I modify the sentence, which will be found on Line 544-545.

One of the critical elements in creating a… investigate new IS is innovation. Line 544-545

  1. The

Ans. Thank you for your suggestion. I modify the sentence, which will be found on Line 582.

These results align with the existing research [41]. Line 582

  1. stay consistent regarding tense use.

Ans. Thank you for your suggestion. I modify the sentence, which will be found on Line 582-583.

These results show that extraverted pupils…better complete creative tasks [38]. Line 582-583

  1. –ed

Ans. Thank you for your suggestion. I modify the sentence, which will be found on Line 618-619.

Higher extroverted students are socially…cautious about selecting a new system. Line 618-619

  1. suppose thorough reordering is required in this section. Conclusion→ Contributions→ limitations→ Suggestions

Ans. Thank you for your suggestion. I reorder the sections of the paper.

Reviewer 2 Report

The Mediating Effects of Students’ Innovativeness and System 2 Usability on their Personality Differences the Recommendation 3 in the E-learning Platform at Post-Pandemic Time

11.   Introduction

p.2 line 83- IS this abbreviation needs explanation...

fig. 1, p. 7: Mark all hypotheses (H1 - H7) listed in the introductory part.

4. Materials and Methods:

4.1 "fundamental level of technological literacy", p.7, line 305 ---explain this as " technological literacy is not ICT or digital literacy...a term that refers to all technologies, not just ICT!

How was the data collected? What e-portal was used and whether it meets all GDPR requirements? Did students give informed consent to participate in the study and how? Is there evidence to support this?

Also outline any ethical considerations.

4.2 Measurement Delphi method

What was the rater agreement or IRR index? What are Cronbach's readings used for? Line 344, p.8 Explain more about the scoring of the items by the raters involved...

All methods used must allow for replicability of the study.

4.3 Procedure

Explain specifically what method/statistical test was used to test each hypothesis. As it is now, only a general description is provided, while there is a lack of consistency in stating the results. Where was AMOS used and what test was performed (e.g., SEM), where SPSS, where was the Hayes macro used, and what model of the Hayes process macro?

5. Results

What was the distribution of the data? Prove this too, as it is crucial for further use of the methods and tests.

5.1

Table 3, it could be that multicollinearity occurs. How was it tested to see if this is the case?

5.2 Reliability and convergent validity

Where is there evidence of convergent validity? What does AVE stand for in Table 2?

No evidence of discriminant validity? Provide this evidence as well.

5.3 Regression analysis

Why is SEM mentioned in the abstract section? Where is the proof of this?

Table 4. what was a reference variable?

Explain this regression analysis used and why, depending on the type of variable and data distribution and quality, and what is the rationale? Was part of SEM or ...

Explain the variables and statistics in the table...

Since both IV and DV were collected from the same respondent, how was common method bias controlled?

Table 5. Which macro model of the Hayes process, if any, was chosen?

Is bootstrapping demonstrably useful and what model/criterion was used?

Results should be reported consistent with the research model.

Generally, a discrepancy is found between theoretical part, applied methodology and results.

Align and correct them accordingly for valid and reliable agreement.

Author Response

  1. Introduction

  1. p.2 line 83- IS this abbreviation needs explanation...

Ans. Thank you for your suggestion. I modify the sentence which will be found on Line 82-83.

It is crucial to consider innovation and system…(IS) usage concerns. Line 82-83

  1. fig. 1, p. 7: Mark all hypotheses (H1 - H7) listed in the introductory part.

Ans. Thank you for your suggestion. I have redrawn figure 1 to make it more easily understandable, which will be found on Page 7.

4. Materials and Methods:

3. 4.1 "fundamental level of technological literacy", p.7, line 305 ---explain this as " technological literacy is not ICT or digital literacy...a term that refers to all technologies, not just ICT!

Ans. Thank you for your suggestion. I have rephrased the sentence to understand it more clearly, which will be found on Lines 309-311.

Second, students should have experience… of ICT devices. Line 309-311

  1. How was the data collected? What e-portal was used and whether it meets all GDPR requirements? Did students give informed consent to participate in the study and how? Is there evidence to support this?

Also outline any ethical considerations.

Ans. Thank you for your suggestion. I have rephrased the sentence to understand it more clearly, which will be found on Lines 316-334.

The target demographic for this… inquiry was students. Line 316

We chose the convenience… with using a randomized sample. [3]. Lines 316-319

All participants were provided with…he current investigation [12,14]. Lines 319-321

An online survey was used to collect… in the study at any time. Lines 321-323

The participants also received a brief… measures implemented [14]. Lines 323-324

The following two elements influenced… the threat to their security. Lines 324-326

The second goal is to establish…security for using participant data. Lines 326-327

The current research did not require… psychological conditions. Lines 328-330

The present study made no use of… during the post-pandemic period. Lines 330-332

The participants chose one of the…based on their understanding. Lines 332-334

4.2 Measurement Delphi method

  1. What was the rater agreement or IRR index? What are Cronbach's readings used for?

Line 344, p.8 Explain more about the scoring of the items by the raters involved...

All methods used must allow for replicability of the study.

Ans. Thank you for your suggestion. I have tried to answer all the issues you rised, which will be found on Lines 389-403.

To test the rater agreement, the Kappa…0.6 is reasonably acceptable. Lines 389-390

The current analysis shows the level… at p<.001 level (Table 4). Lines 390-391

Cronbach's alpha (α) was employed during… of study items. Lines 395-396

It was also calculated as a measure…agreement among the panelists. Lines 396-398

Panelists were asked to provide their… and five is the most suitable. Lines 399-400

Then we calculate the average value…was equal to or higher than 3. Lines 400-402

If the value is lower than 3, it is… item's inclusion for final analysis. Lines 402-403

4.3 Procedure

  1. Explain specifically what method/statistical test was used to test each hypothesis. As it is now, only a general description is provided, while there is a lack of consistency in stating the results. Where was AMOS used and what test was performed (e.g., SEM), where SPSS, where was the Hayes macro used, and what model of the Hayes process macro?

Ans. Thank you for your suggestion. I have tried to answer all the issues you raised, which will be found on Lines 407-412, 414-415, 525-526.

Data analysis was done using… two statistical programs. Lines 407-408

Confirmatory Factor Analysis (CFA)…the reliability using SPSS. Lines 408-410

Data are analyzed using the structural…are tested using AMOS. Lines 410-412

Finally, Hayes’ PROCESS macro was…effect using SPSS [48]. Lines 414-415

Model 4 (Simple mediation) of the… is used for the current analysis. Lines 525-526

5. Results

7. What was the distribution of the data? Prove this too, as it is crucial for further use of the methods and tests.

Ans. Thank you for your suggestion. I have tried to discuss the issue you raised, which will be found on Lines 346-349.

According to Hair et al. [47], if the dataset…the data distribution. Lines 346-347.

Since we have 438 valid responses, the…value is 0.316. Lines 347-348.

Thus, we can reject the alternative hypothesis…distribution. Lines 348-349.

5.1

  1. Table 3, it could be that multicollinearity occurs. How was it tested to see if this is the case?

Ans. Thank you for your suggestion. I have tried to discuss the issue you raised, which will be found on Line 456-459

The Variance inflation factor (VIF) test…to test multicollinearity. Line 456

According to Hair et al. [47], values… higher than 10 indicate high VIF. Line 457

Table 6 shows the VIF of each construct… from 2.137 to 3.258. Lines 457-458

 Thus, we can conclude that no… multicollinearity was seen. Lines 458-459

5.2 Reliability and convergent validity

  1. Where is there evidence of convergent validity? What does AVE stand for in Table 2?

Ans. Thank you for your suggestion. I have tried to discuss the issues you raised, which will be found on Line 463-470

I write the Full-form of AVE, which will be found on Table-5

Convergent validity of the scales is… should be higher than 0.50. Line 463-465

As Table 5 endorses, the factor loading…from 0.72 to 0.83 (Table 5). Line 465-467

AVE value of constructs ranged from… validity is met (Table 5). Line 467-469

  1. No evidence of discriminant validity? Provide this evidence as well.

Ans. Thank you for your suggestion. I have tried to discuss the issue you raised, which will be found on Line 474-478

To test discriminant validity, Hair et al. [47]… constructs in the model. Line 473-475

Table 7 shows the square root of AVE for… for discriminant validity. Line 475-477

5.3 Regression analysis

  1. Why is SEM mentioned in the abstract section? Where is the proof of this?

Ans. Thank you for your suggestion. I have tried to discuss the issue you raised, which will be found on Lines 410-415

Data are analyzed using the structural…are tested using AMOS. Line 410-412

SEM is used since it is a widespread…variable measurement [47]. Line 412-414

Finally, Hayes’ PROCESS macro was…effect using SPSS [48]. Line 414-415

  1. Table 4. what was a reference variable?

Explain this regression analysis used and why, depending on the type of variable and data distribution and quality, and what is the rationale? Was part of SEM or ...

Explain the variables and statistics in the table...

Ans. Thank you for your suggestion. I have tried to discuss the issues you raised, which will be found on Lines 480-487

I have also answered the variables and statistics used in the table, which will be found on Lines 504-505

Regression analysis was conducted…model's relationships. Lines 480-481

It is a consistent method of determining… the post-pandemic time. Lines 481-483

To achieve the purpose of the current… are shown in Table 8. Lines 483-487

EXT= Extraversion; OPE= Openness to experience…adoption  Line 504-505

  1. Since both IV and DV were collected from the same respondent, how was common method bias controlled?

Ans. Thank you for your suggestion. I have tried to discuss the issue you raised, which will be found on Lines 488-490

The common method bias is controlled…  and their measures. Lines 488-490

  1. Table 5. Which macro model of the Hayes process, if any, was chosen?

Ans. Thank you. I have tried answered the issue you raised, which will be found on Lines 525-526.

Model 4 (Simple mediation) of the… is used for the current analysis. Lines 525-526

  1. Is bootstrapping demonstrably useful and what model/criterion was used?

Ans. Thank you for your suggestion. I have tried to answer the issue you raised, which will be found on Lines 491-496

The bootstrapping strategy, which resamples…an efficient alternative. Lines 491-492

Moreover, it enables the creation of… and hypothesis testing. Lines 492-493

By reducing bootstrap estimates of the…the subset of variables. Lines 493-496

  1. Results should be reported consistent with the research model.

Ans. Thank you for your suggestion. I rewrite the Results according to your suggestions, which will be found on Lines 516-543

To validate the scales and assess the…(EFA) was conducted. Line 417-418

The mean value and standard…for each variable are given in Table 5. Line 418-420

Among the personality traits, extroversion…supporting H3a and H3b. Line 436-437

Conscientiousness was not significantly…H3d was not supported. Line 439-441

Thus, H4a, H4b, and H4e were…H4d were also supported. Line 445-447

The Variance inflation factor (VIF) test… to test multicollinearity. Line 455

According to Hair et al. [47], values… higher than 10 indicate high VIF. Line 446

Table 6 shows the VIF of each construct…multicollinearity was seen. Line 456-458

Cronbach’s alpha and CR value of each… reliability and consistency. Line 461-463

Convergent validity of the scales is… should be higher than 0.50. Line 463-465

As Table 5 endorses, the factor loading…from 0.72 to 0.83 (Table 5). Line 465-467

AVE value of constructs ranged from… validity is met (Table 5). Line 467-469

To test discriminant validity, Hair et al. [47]… constructs in the model. Line 473-475

Table 7 shows the square root of AVE for… for discriminant validity. Line 475-477

Regression analysis was conducted to… model's relation-ships. Line 479-480

It is a consistent method of determining… the post-pandemic time. Line 480-482

To achieve the purpose of the current… and system usability. Line 482-484

The common method bias is controlled by… and their measures. Line 487-489

The bootstrapping strategy, which…errors, and hypothesis testing. Line 490-493

By reducing bootstrap estimates of the… the subset of variables. Line 493-495

The results showed that all personality… at the 0.001 level. Line 496-497

However, conscientiousness was…(β = - 0.112; R2 = 0.013). Line 496-499

All personality traits significantly… (β = -0.074; t-value = -1.275). Line 505-506

Similarly, personal innovativeness positively…(β =0.737; R2 = 0.526). Line 510-511

Extroversion, openness, and agreeableness…adoption behavior. Line 515-517

Finally, H6 and H7 were proposed to…and system usability. Line 522-524

Model 4 (Simple mediation) of the…is used for the current analysis. Line 524-525

The findings also demonstrated…and e-learning adoption behavior. Line 535-538

  1. Generally, a discrepancy is found between theoretical part, applied methodology and results.

Align and correct them accordingly for valid and reliable agreement.

Ans. Thank you for your suggestion. I rewrite the theoretical part, applied methodology and results, which will be found on Lines 114-124, 303-415, 516-543

Peng and Dutta [14] explored how different… the pandemic. Line 115-118

Literature employed theories such as… intention [3,10,11,14, 17]. Line 118-121

Despite assertions that the Big Five personality… validity [13,19,20]. Line 121-124

students should have experience using digital…of ICT devices. Line 309-311

The target demographic for this inquiry was students. Line 316

We chose the convenience…with using a randomized sample. [3]. Line 316-319

All participants were provided with…the current investigation [12,14]. Line 319-321

The participants also received a brief…that would be implemented [14]. Line 323-324

The following two elements influenced… the decision to use this tactic. Line 324-325

We first aim to fill in any knowledge gaps…the threat to their security. Line 325-326

The second goal is to establish an appropriate… using participant data. Line 326-327

The current research did not require… or psychological conditions. Line 328-330

The present study made no use… during the post-pandemic period. Line 330-332

The participants chose one of the…based on their understanding. Line 332-334

According to Hair et al. [47], if the dataset…test the data distribution. Line 346-347

Since we have 438 valid responses, the… Shapiro-Wilk test is used. Line 347-348

Table 3 shows the p-value is 0.316. Thus, we… normal distribution. Line 348-349

Every panelist has over ten years…field and receives doctoral degrees. Line 363-364

The final items employed for the current…in Table A1 (Appendix A). Line 372-373

Each factor's mean value, standard deviation, and…from experts. Line 374-375

Cronbach's alpha varied from… 0.778 to 0.946 in this round. Line 375-376

To test the rater agreement, the Kappa Statistic… acceptable. Line 389-390

The current analysis shows the…significant at p<.001 level (Table 4). Line 390-391

Cronbach's alpha (α) was employed…consistency of study items. Line 395-396

It was also calculated as a measure…of agreement among the panelists. Line 396-398

Panelists were asked to provide their… and five is the most suitable. Line 399-400

Then we calculate the average value…was equal to or higher than 3. Line 400-402

If the value is lower than 3, it is…the item's inclusion for final analysis. Line 402-404

Data analysis was done using SPSS and…two statistical programs. Line 407-408

Confirmatory Factor Analysis (CFA) is… scales by AMOS. Line 408-409

Cronbach alpha and composite reliability…the reliability using SPSS. Line 409-410

Data are analyzed using the structural…are tested using AMOS. Line 410-412

Finally, Hayes’ PROCESS macro was…effect using SPSS [48]. Line 414-415

To validate the scales and assess the…(EFA) was conducted. Line 417-418

The mean value and standard…for each variable are given in Table 5. Line 418-420

Among the personality traits, extroversion…supporting H3a and H3b. Line 436-437

Conscientiousness was not significantly…H3d was not supported. Line 439-441

Thus, H4a, H4b, and H4e were…H4d were also supported. Line 445-447

The Variance inflation factor (VIF) test… to test multicollinearity. Line 455

According to Hair et al. [47], values… higher than 10 indicate high VIF. Line 446

Table 6 shows the VIF of each construct…multicollinearity was seen. Line 456-458

Cronbach’s alpha and CR value of each… reliability and consistency. Line 461-463

Convergent validity of the scales is… should be higher than 0.50. Line 463-465

As Table 5 endorses, the factor loading…from 0.72 to 0.83 (Table 5). Line 465-467

AVE value of constructs ranged from… validity is met (Table 5). Line 467-469

To test discriminant validity, Hair et al. [47]… constructs in the model. Line 473-475

Table 7 shows the square root of AVE for… for discriminant validity. Line 475-477

Regression analysis was conducted to… model's relation-ships. Line 479-480

It is a consistent method of determining… the post-pandemic time. Line 480-482

To achieve the purpose of the current… and system usability. Line 482-484

The common method bias is controlled by… and their measures. Line 487-489

The bootstrapping strategy, which…errors, and hypothesis testing. Line 490-493

By reducing bootstrap estimates of the… the subset of variables. Line 493-495

The results showed that all personality… at the 0.001 level. Line 496-497

However, conscientiousness was…(β = - 0.112; R2 = 0.013). Line 496-499

All personality traits significantly… (β = -0.074; t-value = -1.275). Line 505-506

Similarly, personal innovativeness positively…(β =0.737; R2 = 0.526). Line 510-511

Extroversion, openness, and agreeableness…adoption behavior. Line 515-517

Finally, H6 and H7 were proposed to…and system usability. Line 522-524

Model 4 (Simple mediation) of the…is used for the current analysis. Line 524-525

The findings also demonstrated…and e-learning adoption behavior. Line 535-538

Reviewer 3 Report

- Sometimes, it feels like the thought in the sentences were incomplete. For example, no line 35: "The creation of e-learning, a significant information technology (IT) innovation, has made it possible for students and instructors to communicate" leaves you hanging is partially incorrect. Students and instructors had already been communicating even without e-learning.

- The introduction could have been made shorter, yet still somehow misses a few points. It would have been enough to define what e-learning is, at least from this paper's point of view. Then a sentence about how its use has increased in the pandemic, objectively.

- There seems to be a focus on the post-pandemic era, but there is minimal indication that a change in behavior was observed because of the pandemic.

- It was not evident earlier on that the Delphi method was used. I think this is important enough, it should have been mentioned in the abstract.

- The results and discussion's readability can be improved. It is hard to remember what do the Hx values mean.

- When system usability is involved, it is important to mention what is the system the participants are evaluating.

- Aside from Big Five, it was hard to deduce what are the measurement tools (questionnaires) used.

Minor:

- No need to number the results in the abstract

Author Response

  1. - Sometimes, it feels like the thought in the sentences were incomplete. For example, no line 35: "The creation of e-learning, a significant information technology (IT) innovation, has made it possible for students and instructors to communicate" leaves you hanging is partially incorrect. Students and instructors had already been communicating even without e-learning.

Ans. Thank you for your suggestion, I modify the sentence which will be found on Page 1

The creation of e-learning, a significant… whenever and wherever needed [2,3].

2.- The introduction could have been made shorter, yet still somehow misses a few points. It would have been enough to define what e-learning is, at least from this paper's point of view. Then a sentence about how its use has increased in the pandemic, objectively.

Ans. Thank you for your suggestion, I modify the introduction which will be found on Line 44-100.

It offers a platform for exchanging knowledge…broadcasting networks [4,5].

E-learning has received significant notice due…during the epidemic [3,6].

The COVID-19 pandemic forced humans… of education globally [7]. Line 44-46

Almost every country forcibly… students' learning ambitions [2,4]. Line 46-47

Higher educational institutions…continuation of academic activities [3]. Line 47-49

Due to its broad adoption, researchers…in vari-ous sectors [7,8]. Line 50-51

Because of the technological nature…particularly during the pandemic. Line 55-57

This information will explain to…in greater acceptance and usage [7]. Line 57-59

If students' expectations are unmet, convincing…platform is difficult. Line 60-62

The majority of empirical studies…platform adoption [10,11,12]. Line 62-66

The literature claims that a person's personality… making [13,14]. Line 70-72

People with different personalities…to use before the pandemic [2,4]. Line 79-82

Thus, the aim of the study is to…post-pandemic-inspired environments. Line 98-100

  1. - There seems to be a focus on the post-pandemic era, but there is minimal indication that a change in behavior was observed because of the pandemic.

Ans. Thank you for your suggestion. I include several sentences that indicate a change in behavior observed because of the pandemic, which will be found on Line 44-100.

The COVID-19 pandemic forced humans…before the pandemic [3,4]. Line 44-45

People with different personalities will evaluate…before the pandemic [2,4]. Line 79-82

  1. - It was not evident earlier on that the Delphi method was used. I think this is important enough, it should have been mentioned in the abstract.

Ans. Thank you for your suggestion. I mention Delphi method in the abstract.

  1. - The results and discussion's readability can be improved. It is hard to remember what do the Hx values mean.

Ans. Thank you for your suggestion, I modify the Results and Discussion section which will be found on Line 415-626.

To validate the scales and assess the…(EFA) was conducted. Line 417-418

The mean value and standard…for each variable are given in Table 5. Line 418-420

Among the personality traits, extroversion…supporting H3a and H3b. Line 436-437

Conscientiousness was not significantly…H3d was not supported. Line 439-441

Thus, H4a, H4b, and H4e were…H4d were also supported. Line 445-447

The Variance inflation factor (VIF) test… to test multicollinearity. Line 455

According to Hair et al. [47], values… higher than 10 indicate high VIF. Line 446

Table 6 shows the VIF of each construct…multicollinearity was seen. Line 456-458

Cronbach’s alpha and CR value of each… reliability and consistency. Line 461-463

Convergent validity of the scales is… should be higher than 0.50. Line 463-465

As Table 5 endorses, the factor loading…from 0.72 to 0.83 (Table 5). Line 465-467

AVE value of constructs ranged from… validity is met (Table 5). Line 467-469

To test discriminant validity, Hair et al. [47]… constructs in the model. Line 473-475

Table 7 shows the square root of AVE for… for discriminant validity. Line 475-477

Regression analysis was conducted to… model's relation-ships. Line 479-480

It is a consistent method of determining… the post-pandemic time. Line 480-482

To achieve the purpose of the current… and system usability. Line 482-484

The common method bias is controlled by… and their measures. Line 487-489

The bootstrapping strategy, which…errors, and hypothesis testing. Line 490-493

By reducing bootstrap estimates of the… the subset of variables. Line 493-495

The results showed that all personality… at the 0.001 level. Line 496-497

However, conscientiousness was…(β = - 0.112; R2 = 0.013). Line 496-499

All personality traits significantly… (β = -0.074; t-value = -1.275). Line 505-506

Similarly, personal innovativeness positively…(β =0.737; R2 = 0.526). Line 510-511

Extroversion, openness, and agreeableness…adoption behavior. Line 515-517

Finally, H6 and H7 were proposed to…and system usability. Line 522-524

Model 4 (Simple mediation) of the…is used for the current analysis. Line 524-525

The findings also demonstrated…and e-learning adoption behavior. Line 535-538

The current study can be seen as a… accept a new e-learning platform. Line 573-574

Additionally, the research instrument offers… personality variations). Line 574-577

These results show that extraverted…complete creative tasks [38]. Line 582-583

Our analysis identified extroversion as… e-learning platform metrics. Line 589-590

The trait conscientiousness shows…usability of the e-learning platform. Line 594-595

Personal innovativeness and the… and sustainable e-learning adoption. Line 605-607

How-ever, not all personality…usability of the e-learning platform. Line 607-608

The findings of this study urge more effective…consider these facts. Line 612-614

Regarding the negative correlation…condition due to the pandemic. Line 615-617

Stakeholders may develop strategies…use of sustainable e-learning. Line 623-626

The system should be designed with… original and beautiful interfaces. Line 626

  1. - When system usability is involved, it is important to mention what is the system the participants are evaluating.

Ans. Thank you for your suggestion, I modify the phrase which will be found throughout thepaper.1

H1: System usability of e-learning platform… e-learning adoption behavior. Line 170-171

H5a: Extroversion is positively associated… of e-learning platform. Line 272-273

H5b: Openness is significantly associated… of the e-learning plat-form. Line 274-275

H5c: Neuroticism is negatively associated… of the e-learning plat-form. Line 275-276

H5d: Consciousness is positively associated…of the e-learning platform. Line 277-278

H5e: Agreeableness is positively associated… of the e-learning platform. Line 279-280

H7: System usability of e-learning platform…e-learning adoption behavior. Line 292-293

All personality traits significantly affected the…(β = -0.074; t-value = -1.275). Line 496-497

Similarly, personal innovativeness positively…(β =0.737; R2 = 0.526). Line 510-511

The findings also demonstrated that the system…  adoption behavior. Line 535-539

According to our analysis, we identified…the e-learning platform metrics. Line 589-590

The trait conscientiousness shows an insignificant…the e-learning platform. Line 594-595

Personal innovativeness and the system usability…e-learning adoption. Line 605-606

However, not all personality traits proposed by the…e-learning platform. Line 607-608

Regarding the negative correlation between…condition due to the pandemic. Line 615-617

  1. - Aside from the Big Five, it was hard to deduce what are the measurement tools (questionnaires) used.

Ans. Thank you for your suggestion, I include all the questionnaire in the Appendix A (Table A1).

Apart from Big Five, the factors of Personal innovativeness and System usability of the e-learning platform were measured using the responses collected through a questionnaire.

Minor:

  1. - No need to number the results in the abstract

Ans. Thank you for your suggestion, I modify the abstract.

Reviewer 4 Report

The Abstract is written in a clear manner describing in brief the scope of the study, research tools, sample and preliminary data. The Introduction section presents attributes of e learning with a short introduction on personality traits and research questions. In 2.1 the five qualities could be presented in a table format and then analysed. In the Theoretical Bakcground I would expect to see a paragraph with the significance of models as such, like the 5 Factor model. My suggestion would be authors to revise the Theoretical Bakcground section focusing on 5 Factor (and other models perhaps) and then in the Methodology section present their hypotheses regarding their research. The way Hypotheses are presented does not convey a strong, robust picture of the Theoretical Background section and conducted research. In the Results section authors could also discuss the added value of their work not just in terms of their study and research but also in terms of state of the art literature review. In my view the Results and Discussion section need to be more extensive than the Contributions section. The Data Analysis is well presented, though reference on methodology applied could be more extensive and detailed.

Author Response

  1. The Abstract is written in a clear manner describing in brief the scope of the study, research tools, sample and preliminary data.

Ans. Thank you.

  1. The Introduction section presents attributes of e learning with a short introduction on personality traits and research questions.

Ans. Thank you.

  1. In 2.1 the five qualities could be presented in a table format and then analysed.

Ans. Thank you for your suggestion. I rewrite the “Big Five Personality Traits” section and include a Table (Table-1) which will be found on Line 109

  1. In the Theoretical Bakcground I would expect to see a paragraph with the significance of models as such, like the 5 Factor model.

Ans. Thank you for your suggestion. I write a paragraph with the significance of models as such, like the 5 Factor model which will be found on Line 181-124.

Literature employed several theories such as… [3,10,11,14, 17]. Line 118-121

Despite assertions that the Big Five personality…validity [13,19,20]. Line 121-124

  1. My suggestion would be authors to revise the Theoretical Bakcground section focusing on 5 Factor (and other models perhaps) and then in the Methodology section present their hypotheses regarding their research.

Ans. Thank you for your suggestion. I rewrite the theoretical Background and Methodology section which will be found on Line 115-124.

Peng and Dutta [14] explored how different… the pandemic. Line 115-118

Literature employed theories such as… intention [3,10,11,14, 17]. Line 118-121

Despite assertions that the Big Five personality… validity [13,19,20]. Line 121-124

  1. The way Hypotheses are presented does not convey a strong, robust picture of the Theoretical Background section and conducted research.

Ans. Thank you for your suggestion. I rewrite the hypothesis section which will be found on Line 161-294.

Usability is determined by several… and subjective satisfaction [33]. Line 161-163

Personal inventiveness was rationally… behavior [22]. Line 173-174

Strobl et al. [22] discovered that personal…an e-learning system. Line 174-175

Zheng et al. [35] indicated personal…impacts usage intention. Line 175-176

Literature validated relationships between… technologies [35,36]. Line 176-177

Individual innovativeness was defined by…inclination to use them. Line 178-181

Literature indicated that adopting…an extroverted personality [37]. Line 184-185

Extroverted pupils are motivated…and are enthusiastic [38]. Line 185-186

Extroverted students prefer to interact… IT technologies [39]. Line 186-187

Neuroticism negatively impacts…and occupational education [17]. Line 199

Literature explored that the best predictor… is agreeableness [44]. Line 210-212

Literature indicates those with high…develop original ideas [16,17]. Line 216-217

While conscientious people's planning,…stifle creative behavior [42]. Line 239-240

Competence, tenacity, and self-control… successful ideas [45]. Line 240-241

 Aligning with this idea, Rivers [17] found…ability for invention. Line 241-242

Hamilton et al. [41] found high…well people accomplish creative tasks. Line 242-244

Strong extraversion personalities… typically try new things [16]. Line 255

Literature showed that extraversion is…regarding system use [17]. Line 255-256

The extraversion personality trait has…systems and technology [45]. Line 256-258

Conscientious learners are more likely…less conscientious ones [41]. Line 260-261

Diligent students could see the system's…academic achievement [19]. Line 261-263

Agreeable students characteristically have…as valuable partners [19]. Line 268-270

This study considered the mediating processes…adoption behavior. Line 283-284

According to the study's anticipated…ingenuity and system usability. Line 287-289

  1. In the Results section authors could also discuss the added value of their work not just in terms of their study and research but also in terms of state of the art literature review.

Ans. Thank you for your suggestion. I rewrite the result section which will be found on Line 417-538.

 To validate the scales and assess the…(EFA) was conducted. Line 417-418

The mean value and standard…for each variable are given in Table 5. Line 418-420

Among the personality traits, extroversion…supporting H3a and H3b. Line 436-437

Conscientiousness was not significantly…H3d was not supported. Line 439-441

Thus, H4a, H4b, and H4e were…H4d were also supported. Line 445-447

The Variance inflation factor (VIF) test… to test multicollinearity. Line 455

According to Hair et al. [47], values… higher than 10 indicate high VIF. Line 446

Table 6 shows the VIF of each construct…multicollinearity was seen. Line 456-458

Cronbach’s alpha and CR value of each… reliability and consistency. Line 461-463

Convergent validity of the scales is… should be higher than 0.50. Line 463-465

As Table 5 endorses, the factor loading…from 0.72 to 0.83 (Table 5). Line 465-467

AVE value of constructs ranged from… validity is met (Table 5). Line 467-469

To test discriminant validity, Hair et al. [47]… constructs in the model. Line 473-475

Table 7 shows the square root of AVE for… for discriminant validity. Line 475-477

Regression analysis was conducted to… model's relation-ships. Line 479-480

It is a consistent method of determining… the post-pandemic time. Line 480-482

To achieve the purpose of the current… and system usability. Line 482-484

The common method bias is controlled by… and their measures. Line 487-489

The bootstrapping strategy, which…errors, and hypothesis testing. Line 490-493

By reducing bootstrap estimates of the… the subset of variables. Line 493-495

The results showed that all personality… at the 0.001 level. Line 496-497

However, conscientiousness was…(β = - 0.112; R2 = 0.013). Line 496-499

All personality traits significantly… (β = -0.074; t-value = -1.275). Line 505-506

Similarly, personal innovativeness positively…(β =0.737; R2 = 0.526). Line 510-511

Extroversion, openness, and agreeableness…adoption behavior. Line 515-517

Finally, H6 and H7 were proposed to…and system usability. Line 522-524

Model 4 (Simple mediation) of the…is used for the current analysis. Line 524-525

The findings also demonstrated…and e-learning adoption behavior. Line 535-538

  1. In my view the Results and Discussion section need to be more extensive than the Contributions section.

Ans. Thank you for your suggestion, I modify the Results and Discussion section which will be found on Line 415-626.

To validate the scales and assess the…(EFA) was conducted. Line 417-418

The mean value and standard…for each variable are given in Table 5. Line 418-420

Among the personality traits, extroversion…supporting H3a and H3b. Line 436-437

Conscientiousness was not significantly…H3d was not supported. Line 439-441

Thus, H4a, H4b, and H4e were…H4d were also supported. Line 445-447

The Variance inflation factor (VIF) test… to test multicollinearity. Line 455

According to Hair et al. [47], values… higher than 10 indicate high VIF. Line 446

Table 6 shows the VIF of each construct…multicollinearity was seen. Line 456-458

Cronbach’s alpha and CR value of each… reliability and consistency. Line 461-463

Convergent validity of the scales is… should be higher than 0.50. Line 463-465

As Table 5 endorses, the factor loading…from 0.72 to 0.83 (Table 5). Line 465-467

AVE value of constructs ranged from… validity is met (Table 5). Line 467-469

To test discriminant validity, Hair et al. [47]… constructs in the model. Line 473-475

Table 7 shows the square root of AVE for… for discriminant validity. Line 475-477

Regression analysis was conducted to… model's relation-ships. Line 479-480

It is a consistent method of determining… the post-pandemic time. Line 480-482

To achieve the purpose of the current… and system usability. Line 482-484

The common method bias is controlled by… and their measures. Line 487-489

The bootstrapping strategy, which…errors, and hypothesis testing. Line 490-493

By reducing bootstrap estimates of the… the subset of variables. Line 493-495

The results showed that all personality… at the 0.001 level. Line 496-497

However, conscientiousness was…(β = - 0.112; R2 = 0.013). Line 496-499

All personality traits significantly… (β = -0.074; t-value = -1.275). Line 505-506

Similarly, personal innovativeness positively…(β =0.737; R2 = 0.526). Line 510-511

Extroversion, openness, and agreeableness…adoption behavior. Line 515-517

Finally, H6 and H7 were proposed to…and system usability. Line 522-524

Model 4 (Simple mediation) of the…is used for the current analysis. Line 524-525

The findings also demonstrated…and e-learning adoption behavior. Line 535-538

The current study can be seen as a… accept a new e-learning platform. Line 573-574

Additionally, the research instrument offers… personality variations). Line 574-577

These results show that extraverted…complete creative tasks [38]. Line 582-583

Our analysis identified extroversion as… e-learning platform metrics. Line 589-590

The trait conscientiousness shows…usability of the e-learning platform. Line 594-595

Personal innovativeness and the… and sustainable e-learning adoption. Line 605-607

How-ever, not all personality…usability of the e-learning platform. Line 607-608

The findings of this study urge more effective…consider these facts. Line 612-614

Regarding the negative correlation…condition due to the pandemic. Line 615-617

Stakeholders may develop strategies…use of sustainable e-learning. Line 623-626

The system should be designed with… original and beautiful interfaces. Line 626

  1. The Data Analysis is well presented, though reference on methodology applied could be more extensive and detailed.

Ans. Thank you for your suggestion, I include references which will be found on Line 319-321, 323-324.

All participants were provided with…the current investigation [12,14]. Line 319-321

The participants also received a brief…that would be implemented [14]. Line 323-324

Round 2

Reviewer 2 Report

The authors have done a great job in improving and enhancing the manuscript. In its present form, it brings some new advances and added value to the field.

Some minor points still need to be improved:

p.8. 328: Give a better solution to this claim about the approval of the study, since some critical personal data were collected: Gender, age, education level... !!!

 Table 3, p.8. l.350: Give the results of the Shapiro-Wilk test for all constructs involved in the study! see table 5 where you have all constructs!

Author Response

  1. The authors have done a great job in improving and enhancing the manuscript. In its present form, it brings some new advances and added value to the field.

Ans. Thank you for your observation.

Some minor points still need to be improved:

  1. p.8. 328: Give a better solution to this claim about the approval of the study, since some critical personal data were collected: Gender, age, education level... !!!

Ans. Thank you for your suggestion. I rewrite the paragraph and try to explain why ethical consideration was not taken for the current study, which will be found on Line 329-340

The current research has not taken…makeup, or psychological conditions. Line 329-331

Additionally, no laboratory results…platform during the post-pandemic time. Line 331-334

Based on their knowledge, participants… strongly disagree to strongly agree). Line 334-335

Also, we briefly explained the overall… It was selected for meeting two issues. Line 335-337

First, to remove any… opinion about the system's potential future uses. Line 337-339

Respondents were also… to be pulled out of participation during the survey. Line 339-340

  1. Table 3, p.8. l.350: Give the results of the Shapiro-Wilk test for all constructs involved in the study! see table 5 where you have all constructs!

Ans. Thank you for your suggestion. I include all the information in Table 3, which will be found on Line 357.

Reviewer 3 Report

Thanks for your hard work in addressing all the reviewers' comments. I particularly appreciate you adding proper validation checks to the questionnaire you created. I still recommend you remove the pandemic-related items. Even though the pandemic may have mattered, there is still no evidence as to how your results are affected by the pandemic, may it be directly (e.g., compare results before and after the pandemic) or indirectly (e.g., experts in the Delphi sessions explicitly asked for their thoughts about pandemic effects). Not everything should be related to the pandemic. If you insist, it could be presented as future work with justification for why it is worthy of additional labor.

Author Response

  1. Thanks for your hard work in addressing all the reviewers' comments. I particularly appreciate you adding proper validation checks to the questionnaire you created.

I still recommend you remove the pandemic-related items. Even though the pandemic may have mattered, there is still no evidence as to how your results are affected by the pandemic, may it be directly (e.g., compare results before and after the pandemic) or indirectly (e.g., experts in the Delphi sessions explicitly asked for their thoughts about pandemic effects). Not everything should be related to the pandemic. If you insist, it could be presented as future work with justification for why it is worthy of additional labor.

Ans. Thank you for your comment. I try to modify the sentences, which could be found in the questionnaire (Personal innovativeness PI1, PI2, and PI4), 79-82, 116-119, 139-141, 331-334, 567-569.

People with different personalities… so favorable to use before the pandemic [2,4]. Line 79-82

Peng and Dutta [14] explored how…E-learning system during the pandemic era. Line 116-119

The impact of innovation on students' thoughts… especially in the post-pandemic time. Line 139-141

Additionally, no laboratory results…during the post-pandemic time. Line 331-334

It is rational to presume less agreeable students…even in the post-pandemic time. Line 567-569